

# Cloud-based configurable data stream processing architecture in rural economic development

Haohao Chen[1] and Fadi Al-Turjman[2]

[1] College of Management, Wuhan Technology and Business University, Wuhan, Hubei, China
[2] Artificial Intelligence, Software, and Information Systems Engineering Departments, Research Center for AI and IoT, AI and Robotics Institute, Near East University, Nicosia, Turkey

## ABSTRACT

**Purpose:** This study aims to address the limitations of traditional data processing methods in predicting agricultural product prices, which is essential for advancing rural informatization to enhance agricultural efficiency and support rural economic growth.

**Methodology:** The RL-CNN-GRU framework combines reinforcement learning (RL), convolutional neural network (CNN), and gated recurrent unit (GRU) to improve agricultural price predictions using multidimensional time series data, including historical prices, weather, soil conditions, and other influencing factors. Initially, the model employs a 1D-CNN for feature extraction, followed by GRUs to capture temporal patterns in the data. Reinforcement learning further optimizes the model, enhancing the analysis and accuracy of multidimensional data inputs for more reliable price predictions.

**Results:** Testing on public and proprietary datasets shows that the RL-CNN-GRU framework significantly outperforms traditional models in predicting prices, with lower mean squared error (MSE) and mean absolute error (MAE) metrics.

**Conclusion:** The RL-CNN-GRU framework contributes to rural informatization by offering a more accurate prediction tool, thereby supporting improved decision-making in agricultural processes and fostering rural economic development.

# INTRODUCTION

Rural economic development has garnered significant attention from governments and organizations worldwide recently. Efforts have been made to bridge the gap between rural and urban economies, focusing on enhancing agricultural productivity, improving infrastructure, and expanding market access. However, rural regions face numerous challenges, including limited access to modern technology, a shortage of skilled labor, and inadequate infrastructure, which hinder economic development (*Tukhtaboev, 2023*). In this context, rural informatization has emerged as a powerful tool to address these challenges and promote sustainable development. Rural informatization enhances agricultural practices by integrating advanced technologies such as the Internet of Things (IoT) and big data analytics. These technologies allow farmers to make decisions regarding

Corresponding author
Haohao Chen,
20160000139@wtbu.edu.cn

crop management and resource allocation, thereby increasing productivity and reducing environmental impact (*Lin et al., 2023*). Cloud computing platforms are crucial in advancing rural informatization by offering the necessary computing power and storage resources to manage and analyze extensive agricultural data. These platforms perform complex analytics and generate real-time insights that drive informed decision-making in agricultural practices. One of the key benefits of cloud platforms is their scalability and cost-efficiency, allowing farmers to seamlessly access, process, and interpret data from diverse sources such as sensors, satellites, and weather monitoring systems. Moreover, these platforms enable efficient data integration and sharing, promoting collaboration among farmers, researchers, agricultural consultants, and policymakers. This interconnected network enhances agricultural productivity by enabling predictive analytics for crop management, optimizing resource allocation, and facilitating the development of precision agriculture techniques tailored to local conditions. This collaborative approach fosters innovation and accelerates the dissemination of rural informatization practices (*Morchid et al., 2024*). Therefore, integrating cloud computing platforms and fully utilizing advanced science and technology are crucial for promoting rural economic development at this stage.

The development of rural informatization using cloud computing platforms presents significant potential for enhancing agricultural productivity and ensuring food security. By harnessing the power of cloud computing, vast amounts of agricultural data can be collected, stored, and analyzed to provide invaluable insights to farmers. These insights facilitate the prediction of crop yields, optimization of resource utilization, and real-time monitoring of crop health (*Liu et al., 2023*). Machine learning (ML) and deep learning (DL) technologies are pivotal in analyzing this data and generating precise predictions. Data collected by IoT sensors in the field can be transmitted to the cloud, which sophisticated algorithms process and analyze. Machine learning techniques such as support vector machines (SVM) and Random Forests are employed for crop pest and disease detection and classification, analyzing image data and sensor inputs to ascertain crop health. Techniques such as decision tree and gradient boosting machine (GBM) are utilized for yield prediction and risk assessment, aiding farmers and agricultural managers in making informed decisions (*Dawn et al., 2023*). Deep learning methods, including CNN, are highly effective in image processing and feature extraction, making them suitable for analyzing remote sensing images and classifying hyperspectral images (*Singh & Khan, 2023*). Cloud platforms offer vital advantages in processing large-scale agricultural data, such as scalability, real-time analysis, and integration capabilities. These platforms can efficiently handle vast datasets from sources like satellites, IoT sensors, and weather stations, enabling accurate and timely agricultural forecasts. Cloud technologies facilitate predictive modeling for crop yields, pest outbreaks, and resource management by providing tools like distributed computing and machine learning frameworks. Additionally, the centralized nature of cloud platforms allows for seamless collaboration among farmers, researchers, and policymakers, improving decision-making and optimizing agricultural productivity. This scalability and efficiency make cloud platforms invaluable for modern agrarian forecasting.

The synergy between cloud computing platforms and deep learning methods enhances the accuracy of agricultural forecasting and significantly fosters rural economic development. Firstly, precise agricultural price and yield forecasts empower farmers to devise scientific planting and sales strategies, mitigating market risks and enhancing economic returns. Secondly, cloud computing platforms are crucial in data processing and storage, providing a robust data foundation for model training and optimization through efficient cloud data management. By facilitating more effective data analysis and management, these platforms contribute to the intense and rapid development of the rural economy. The specific contributions of this article are as follows:

(1) A forecasting framework incorporating multidimensional influencing factors is proposed to address the challenge of predicting agricultural commodity prices within the context of rural economic development.

(2) A *reinforcement learning (RL), convolutional neural network (CNN), and gated recurrent unit (GRU) framework (RL-CNN-GRU),* integrating reinforcement learning and deep learning, is proposed to manage agricultural data's complexity and time-series dependence.

(3) The RL-CNN-GRU model significantly outperforms traditional methods such as CNN and LSTM in price prediction tests on public and proprietary datasets, particularly in metrics such as mean squared error (MSE) and mean absolute error (MAE).

The article is arranged as follows: "Related Works" reviews related works on cloud computing in intelligent agriculture and the prediction of agricultural product information. "Methodology" details the establishment of the proposed RL-CNN-GRU framework. "Experiment Result and Analysis" presents the experimental results and practical tests. "Discussion" provides a discussion, and the conclusion is drawn at the end.

## RELATED WORKS

### Cloud computing and rural informatization

The cloud computing platform aids rural economic development by providing efficient data processing and analysis capabilities. It collects and analyzes agricultural data in real-time to optimize resource allocation and improve production efficiency. Additionally, cloud computing supports market price prediction, assisting farmers in developing better sales strategies and reducing losses (*Cinar & Bharadiya, 2023*). The flexibility and scalability of cloud services also enable rural areas to adapt to technological advances and achieve sustainable development swiftly. Amazon's primary offerings (*Mark & Bommu, 2024*) provide computing and storage services to organizations with storage servers, bandwidth, and CPU resources. Storage servers and bandwidth are charged according to capacity, and CPU is charged based on computing time, with options for monthly fees. The orderly development of foreign cloud computing platforms, specifically for agricultural industry information platforms, ensures a perfect organizational system that aligns research with actual needs (*Vellela et al., 2023*). These customized, innovative agriculture systems facilitate large-scale coordinated agricultural production models using drones, sensors, detectors, control systems, and other means for targeted data collection and automated control. *Ammad Uddin et al. (2021)* implemented a rural agricultural

information system using edge computing to rapidly address challenging locations to deploy edge facilities. *Raghuvanshi et al. (2022)* proposed smart irrigation applications combining machine learning with extensive data mining and neural networks, suggesting that cloud computing will further the development of smart agriculture. *Oteyo et al. (2021)*, through research on smart agriculture systems and cloud computing, provided direction for the future development of agriculture. *Gunjan et al. (2021)* emphasized that traditional agriculture in India must evolve towards science and intelligence, focusing on smart agriculture to improve current agricultural backwardness through modern information technology. *Gouraud (2014)* highlighted the importance of information sharing in informatized agricultural data, proposing that agricultural community communication should leverage rapidly developing information technology for knowledge sharing, which is crucial for regional or national agricultural development. The combination of agricultural production and information technology is highly prioritized in the United States. Since as early as 2008, the U.S. has integrated cloud computing with agricultural information platforms, smart agriculture systems, GIS, remote sensing, database technology, and multimedia, maximizing information utilization through national-level data centers and data-sharing platforms to centralize, standardize, and normalize agricultural data (*Mesías-Ruiz et al., 2023*). To address this, integrating case studies such as using Amazon Web Services (AWS) in smart agriculture can be effective. For example, AWS IoT solutions have been applied in precision farming to monitor soil moisture levels and optimize irrigation, demonstrating the tangible benefits of cloud platforms in enhancing agricultural productivity.

## Rural economic intelligence forecasting study

In "Cloud computing and rural informatization", we observe that with the continuous advancement of rural informatization, people utilize more intelligent methods to predict agricultural product price trends and other information, achieving more precise data analysis and facilitating the digital transformation of the rural economy. Various factors influence the price of agricultural products, and multiple models and methods can be employed to analyze these factors. Influencing factors include random events, residential consumption, production logistics, the level of digital agriculture infrastructure, economic policy, weather conditions, and more. *Wawale et al. (2022)* employed the Autoregressive Integrated Moving Average (ARIMA) model to predict rice prices in India from 2016 to 2021, using MSE and mean absolute percentage error (MAPE) to test the accuracy of the forecasts. *Montecillo (2021)* constructed ARIMA, vector autoregressive (VAR), and vector error correction (VEC) models for corn futures prices in Mexico and the U.S. *Gu et al. (2022)* proposed a dual input attention long short-term memory (DIA-LSTM) framework for effective prediction of monthly prices of cabbage and radish. *Ye et al. (2021)* introduced an Heterogeneous Graph-enhanced LSTM (HGLTSM) model to predict weekly hog prices. Historical prices of essential agricultural commodities from recent years were extracted, and discussions from a specialized online community were used to construct the heterogeneous graph. *Mahto et al. (2021)* utilized artificial neural networks for short-term forecasting of price data for sunflower and soybean seeds in the markets of Maharashtra

and Andhra Pradesh, India, comparing the results with those of a conventional ARIMA model.

The study above demonstrates that cloud computing platforms have significantly advanced the rural economy by providing efficient data processing and analytical capabilities. Cloud computing can collect and analyze vast amounts of agricultural data in real-time, optimize resource allocation, enhance production efficiency, and assist farmers in developing superior sales strategies and minimizing losses through market price forecasts. Additionally, the flexibility and scalability of cloud computing allow rural areas to adapt to technological advances and achieve sustainable development swiftly.

However, more than relying solely on traditional data analysis methods is required for further improvements in the agricultural economy. With the ongoing development of rural informatization, more intelligent methods are necessary to predict agricultural price trends and sales. Modern research indicates that techniques combining extensive data mining and machine learning, such as ARIMA models, LSTM networks, and artificial neural networks, can more accurately forecast agricultural prices and demand. These advanced prediction methods can identify multiple factors affecting prices and provide more precise and reliable decision support, aiding farmers in optimizing their production and sales strategies, stabilizing the supply and demand of agricultural products, and ultimately fostering better development of the agricultural economy.

## METHODOLOGY

### CNN and GRU

CNN can be employed for image processing and effective feature extraction from multidimensional time series data. These data comprise sequences of multiple variables over time, from which CNNs can discern significant patterns and features due to their robust feature extraction capabilities. In these datasets, each dimension represents a distinct feature or variable; for example, agricultural data may include multiple time series such as temperature, humidity, and rainfall (*Shi et al., 2022*).

CNNs can capture the complex relationships and temporal dependencies between these variables through the operations of convolutional and pooling layers. This article uses one-dimensional convolution (1D convolution) to process time series data, where the convolutional kernel slides along the time dimension, extracting various time-dependent features through different convolutional kernels. The convolution operation is mathematically represented in Eq. (1):

$$(I * K)(t) = \sum_{\tau} I(t - \tau) K(\tau) \tag{1}$$

where $I$ is the input sequence, $K$ is the convolution kernel, and $t$ is the time step. The convolutional kernel acts as a sliding window in the convolution operation, capturing local features as it moves over the input data step by step. In time-series data processing, the kernel moves along the time steps, computing the weighted sum between the input data and the kernel at each point to extract patterns and features from the time series. This process can be seen as applying a weighted filter within a small range to extract localized

features, thereby capturing local trends and changes in the time series data. In this way, the convolutional kernel effectively identifies patterns, such as periodic fluctuations or sudden changes, providing helpful information for subsequent feature extraction and analysis.

Apply a nonlinear activation function (*e.g.*, ReLU) to the output of the convolutional layer, thereby enhancing the model's expressive power. Utilize pooling operations (*e.g.*, maximum or average pooling) to reduce the temporal dimension, retaining crucial features while mitigating computation and overfitting risks. Progressively extract higher-level features by stacking multiple convolutional and pooling layers, capturing intricate time-dependent patterns. Ultimately, the extracted features are unfolded through the fully connected layers, preparing them for the final feature representation or classification task. Following feature extraction *via* 1DCNN, a GRU layer is appended further to enhance the performance of the time series classification method.

The GRU is a variant of the RNN designed to address the issues of gradient vanishing and gradient explosion in traditional RNNs when processing long sequence data by introducing a gate mechanism. GRUs simplify the structure of LSTMs by having fewer parameters while effectively capturing long-term dependencies in sequence data. Its primary structural components include update gates and reset gates, with update gates regulating the influence of previous memory on the current state and reset gates determining the extent of past information to discard (*Cahuantzi, Chen & Güttel, 2023*). The reset gate controls the incorporation of new input with past memory by deciding whether to ignore past states or not. When the reset gate is near zero, the GRU effectively forgets the past state, making it suitable for capturing short-term dependencies. The update gate, on the other hand, determines how much of the past state (long-term memory) should be carried forward to the next state. It helps maintain long-term dependencies by controlling how much of the previous state remains in the current state. When the update gate is activated, the GRU retains more information from the past, making it useful for sequences where long-term memory is crucial. This dual mechanism allows GRUs to efficiently manage memory efficiently, leading to faster training and reduced complexity compared to LSTM units, which use three gates instead of two.

$$r_t = \sigma(W_r \cdot [h_{t-1}, x_t] + b_r) \tag{2}$$

where $r_t$ is the output of the reset gate, $W_r$ and $b_r$ are the weights and biases, $\sigma$ is the Sigmoid activation function, $h_{t-1}$ is the hidden state of the previous moment, and $x_t$ is the input of the current moment. The update gate is then shown in Eq. (3):

$$z_t = \sigma(W_z \cdot [h_{t-1}, x_t] + b_z) \tag{3}$$

where $z_t$ is the output of the update gate, $W_z$ and $b$ are the weights and biases? According to the output of the update gate $z_t$ and the candidate hidden state $\tilde{h}_t$, update the hidden state $h_t$ at the current moment. The GRU effectively addresses the issue of gradient vanishing in traditional RNNs when processing long sequence data through its reset and update gate mechanisms. Its relatively simple structure and high computational efficiency extract valuable features from multidimensional time series data, providing robust support for classification or regression tasks. Consequently, this article proposes the integration of

CNN and GRU networks to perform comprehensive data feature extraction, thereby enhancing the performance of the features and ultimately improving the final prediction outcomes.

Combining CNN and GRU can be further justified by emphasizing that 1D-CNNs are particularly efficient for extracting local features and patterns from time-series data as they apply convolutional operations along the temporal dimension. This enables them to capture short-term dependencies effectively, which is crucial for tasks involving sequential data. In contrast, models like Transformers and attention-based mechanisms are more suited for capturing long-range dependencies but may come with higher computational costs and complexity. To strengthen the argument, experiments comparing the performance of 1D-CNN and GRU against Transformer-based models on the same time-series dataset could provide empirical evidence, demonstrating the advantages of the chosen architecture regarding accuracy, efficiency, and model interpretability.

## Reinforcement learning DDPG

Deep Deterministic Policy Gradient (DDPG) is a reinforcement learning algorithm for high-dimensional continuous action spaces. It merges the strengths of Deep Q-Network (DQN) and policy gradient methods. It effectively manages high-dimensional continuous action tasks using neural networks to approximate the policy and Q-value functions. In the DDPG method, the main networks include the policy network and the Q-value network, the deterministic policy $\mu(s|\theta^{\mu})$, which outputs the actions $s$ in a given state $a$ (*Islam, Ball & Goodin, 2024*). The Q-value $Q(s, a|\theta^{Q})$ for a given state $s$ and action $a$ is also estimated. Its overall structure is shown in Fig. 1:

The Critic network loss function, *i.e.*, the loss between Q and Q′, for both the policy network and the target network for a given graph, is computed by the following equation:

$$L(\theta^{Q}) = \mathrm{E}\left[(y_t - Q(s_t, a_t|\theta^{Q}))^2\right] \tag{4}$$

In this deep reinforcement learning framework, network and parameter initialization is required to initialize the policy $\mu(s|\theta^{\mu})$and Q-value network $Q(s, a|\theta^{Q})$ s and randomly initialize the parameters $\theta^{\mu}\theta^{Q}$. Initialize the target network $\mu'$ and Q′ and set the parameters to the same as the primary network, *i.e.*, $\theta^{\mu'} \leftarrow \theta^{\mu}\theta^{Q'} \leftarrow \theta^{Q}$. Execute the action $a_t = \mu(s_t|\theta^{\mu}) + \mathrm{N}_t$ in the environment, where $\mathrm{N}_t$ is exploring the noise. During the updating process of Q′ network, we need to calculate the target value as follows:

$$y_i = r_i + \gamma Q'(s_{i+1}, \mu'(s_{i+1}|\theta^{\mu'})|\theta^{Q'}) \tag{5}$$

With the completion of the update of the policy network, the update of the target network is further realized by employing a soft update, and the update process is as follows:

$$\theta^{Q'} \leftarrow \tau\theta^{Q} + (1 - \tau)\theta^{Q'} \tag{6}$$

$$\theta^{\mu'} \leftarrow \tau\theta^{\mu} + (1 - \tau)\theta^{\mu'} \tag{7}$$

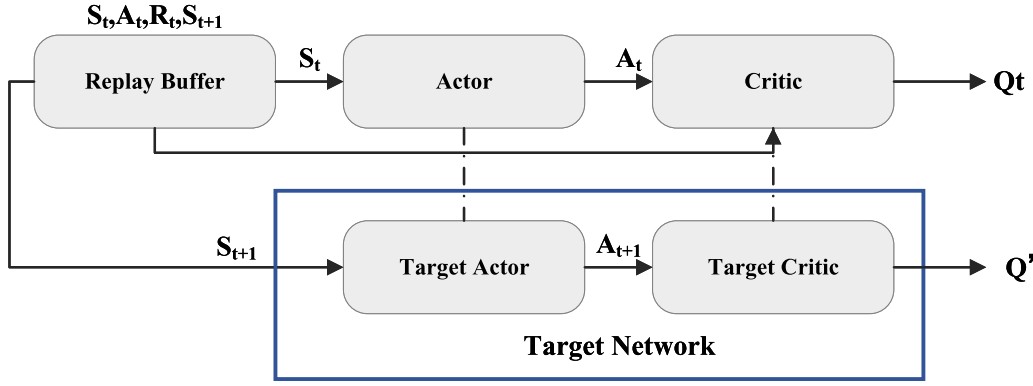

**Figure 1  The update framework for DDPG.**     

$\tau$ a small weighting factor balances the relationship between the coefficients and thus controls the convergence rate.

## The establishment of the RL-CNN-GRU for the price prediction

Upon completing the foundational feature extraction with CNN and GRU, we developed an RL-CNN-GRU network framework, integrating reinforcement learning for the analysis and prediction of agricultural product prices in the context of rural economic development and smart agriculture. This framework initially employs CNN to perform feature extraction on multidimensional time series data, capturing critical spatial features. Subsequently, GRU is utilized to identify time-dependent relationships and long-term dependencies within the series data. Finally, integrating a reinforcement learning strategy optimizes the price prediction model, enhancing its accuracy and robustness. The overall structure of the framework is illustrated in Fig. 2.

According to Fig. 2, the framework initially utilizes time series data, such as historical prices and discrete influences, as feature inputs to create input vectors for the CNN network's feature extraction. This article employs a 1DCNN structure, incorporating two convolutional layers and two pooling layers to enhance model feature performance. Following the feature extraction by the convolutional layers, the GRU network furthers the features to strengthen the time series features. Finally, the model is optimized through a reinforcement learning strategy to enhance the accuracy and robustness of the predictions. This integrated approach aims to improve the precision of agricultural product price forecasts, thereby providing a scientific basis and technical support for the advancement of smart agriculture.

## EXPERIMENT RESULT AND ANALYSIS

### Dataset and experiment setup

Upon completing the model construction, we proceed to analyze the model using an appropriate dataset. Given the application context and the prediction needs for agricultural products, we selected the Global Yield Gap Atlas (*Fayazi et al., 2023*) dataset (https://zenodo.org/records/8280551, doi: 10.5281/zenodo.8280551). The Global Yield Gap Atlas (GYGA) is a comprehensive agronomic database that provides data on up to 13 major food

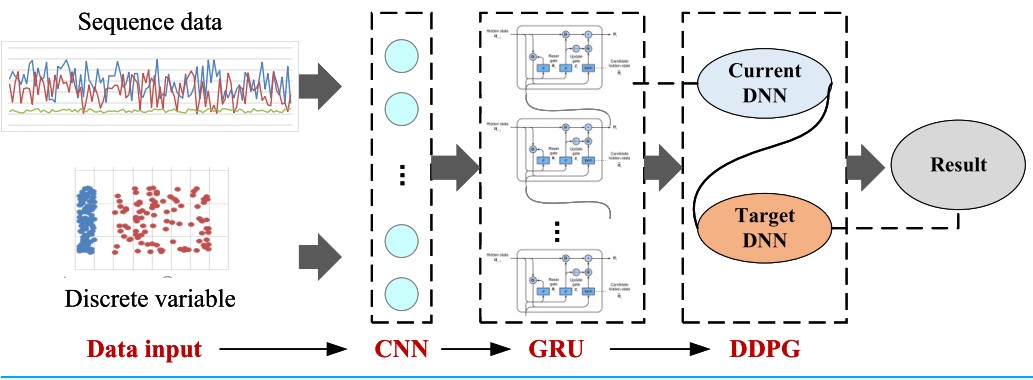

**Figure 2** Framework for the RL-CNN-GRU.

crops across 70 countries and six continents. It includes information on actual and potential crop yields, yield gaps, water productivity, nutrient requirements, and other relevant agronomic factors such as weather, soil, and crop management systems. This dataset offers comprehensive global agricultural data, highlighting the disparities between potential and actual yields of various crops under different climatic and soil conditions. As a collaborative initiative of international agricultural research institutions, the project aims to assist policymakers, researchers, and farmers in optimizing agricultural production, reducing yield gaps, and enhancing food security through scientific data and analysis. The corresponding yields and their calculated average yields for different regions are illustrated in Fig. 3.

Where YW and YP denote tons per harvested hectare at standard moisture content, after introducing the public dataset, we proceeded with the training and testing of the model, utilizing the following experimental environment as shown in Table 1.

In the process of method comparison, we utilize MSE and MAE indices to evaluate the overall performance of the proposed method. Various fundamental methods are selected for model comparison, given the predictive regression nature based on multi-source data. This article includes multiple time series analysis prediction methods, such as single CNN, LSTM, and GRU, and combined frameworks like CNN-LSTM (*Kim & Cho, 2019*) and CNN-GRU (*Sajjad et al., 2020*). This approach aims to achieve a more accurate evaluation of model performance. For the model's parameters, we added two pooling layers after the two convolutional layers in the convolutional layer, with a kernel size of 3 and a GRU of 64 units in two layers. The Actor and Critic learning rates in DDPG reinforcement learning were 0.0001 and 0.001, respectively. The model was trained on 50 epochs using the Adam optimizer.

## Model comparison and result analysis

We have compared the performance of the models using the public dataset and employed MSE and MAE metrics to evaluate and compare the performance of various methods. Figure 4 presents the final values of the loss function changes and MSE observed during training.

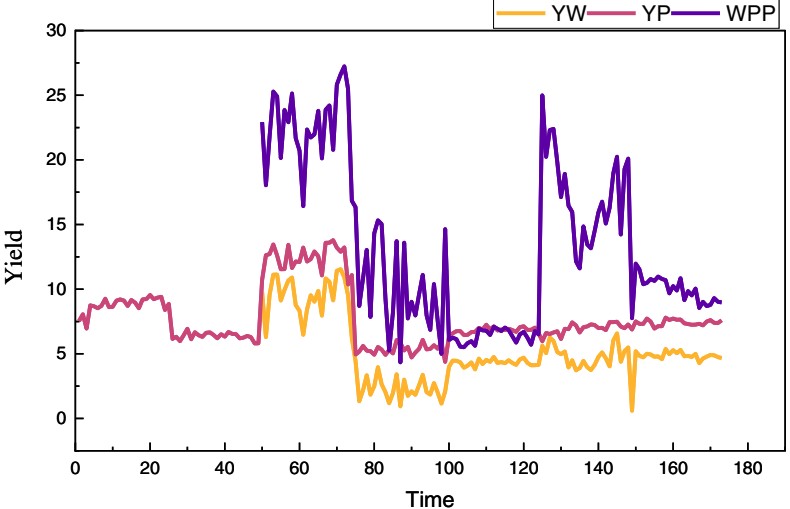

**Figure 3  The data from the public dataset.**

In Fig. 4, we observe that the initial loss values of the models are similar, attributable to the relatively small amount of data and the smoother trend. For the RL-CNN-GRU proposed, integrating the reinforcement learning module improves dynamic interaction, allowing quicker convergence and iteration. Compared to other methods, it demonstrates a faster and smoother overall iteration process. The final MSE value for the proposed framework is 1.91, significantly outperforming the 2.12 observed without the reinforcement learning module and other traditional methods. To provide a more intuitive analysis of the model's performance, we calculated and compared its MAE and overall computation time with the results depicted in Table 2 and Fig. 5.

In Fig. 5, since the MAE metric does not involve the calculation of the square, its overall value is slightly lower than that of the MSE metric. The MAE of the proposed method is 1.87, still outperforming other methods. Additionally, we recorded the training times of different models, as detailed in Table 2. The proposed method's training time exceeds 13 min due to the addition of the reinforcement learning module, which results in a longer runtime compared to traditional methods like CNN-GRU. However, the rapid convergence capability of the reinforcement learning module minimizes this time difference. In smart agriculture, ensuring a better fitting performance is more critical. Furthermore, in terms of traditional methods, the GRU model significantly improves the computational speed by using only two gates for control, making it noticeably faster than the LSTM method. Therefore, integrating reinforcement learning with multiple models effectively meets the yield prediction requirements in smart agriculture.

## Cloud platform data collection and model testing

For cloud platform data, this article utilizes OpenStack, an open-source cloud computing platform, to analyze regional agricultural data. OpenStack comprises a suite of interconnected services that manage and control extensive computing, storage, and networking resources, which are crucial for storing and managing various collected data.

Table 1 The experiment environment information.

| Environment | Information |
| --- | --- |
| CPU | I5-13500 |
| GPUs | RTX 2080 |
| Language | Python 3.5 |
| Framework | Tensorflow |

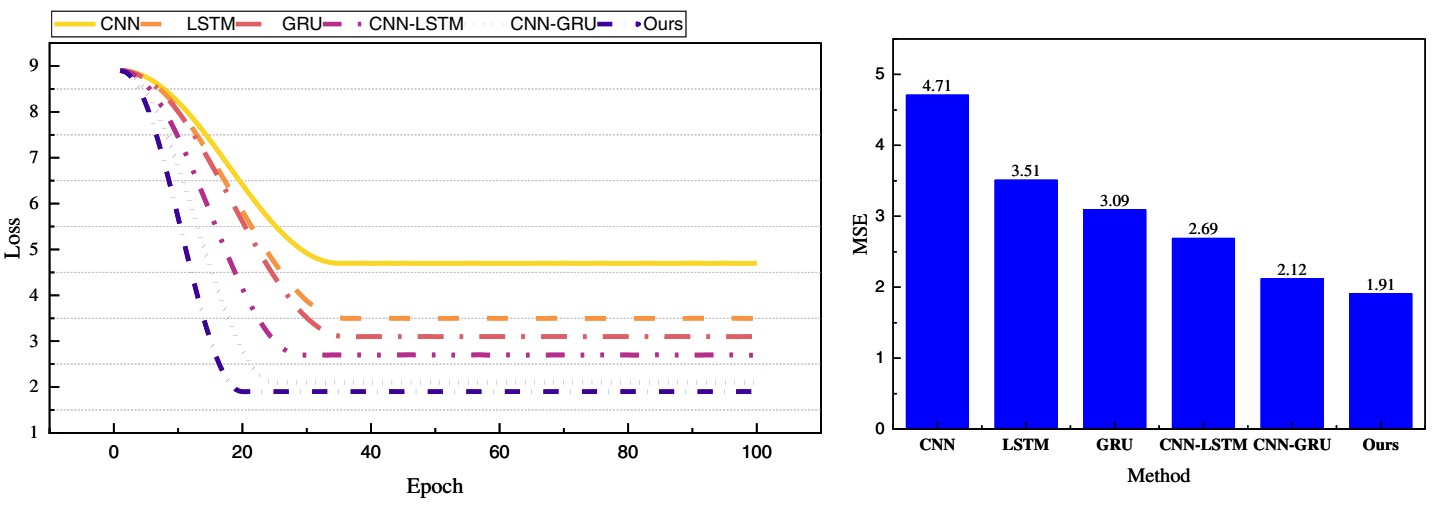

Figure 4 Training loss and MSE on public dataset.

Agricultural production data encompasses soil data (*e.g.*, soil moisture, temperature, pH, and nutrient content), meteorological data (*e.g.*, temperature, humidity, rainfall, and wind speed), crop growth data (*e.g.*, growth stage, leaf area index, pest and disease information), and agricultural machinery operations data (*e.g.*, time of use, location, area of operation, and fuel consumption). Timely data collection and storage are essential for effective analysis and decision-making. To test the model, this study collected data from a regional smart agriculture production area over 100 observation periods for over 10 crops. The collected information includes Soil Moisture, Soil Temperature, Soil pH, Nutrient Content, Temperature, Humidity, Irrigation, Wind Speed, and Product Price. An example of a row of various types of information at the observation time is presented in Fig. 6.

After completing data collection, denoising, and outlier removal, we analyzed the daily price prediction and production on the corresponding date. As previously described, the model initialization was performed by migrating model parameters. Based on this, we divided the data into a training set and a test set with the classic ratio, *i.e.*, 70% for training and 30% for test. The corresponding results are displayed in Figs. 7 and 8.

In Fig. 7, we present the corresponding price prediction data. The overall trend is relatively smooth, indicating a good fit for the price fluctuation characteristics and accurately capturing the general upward trend. The MSE and MAE are 0.19 and 0.15,

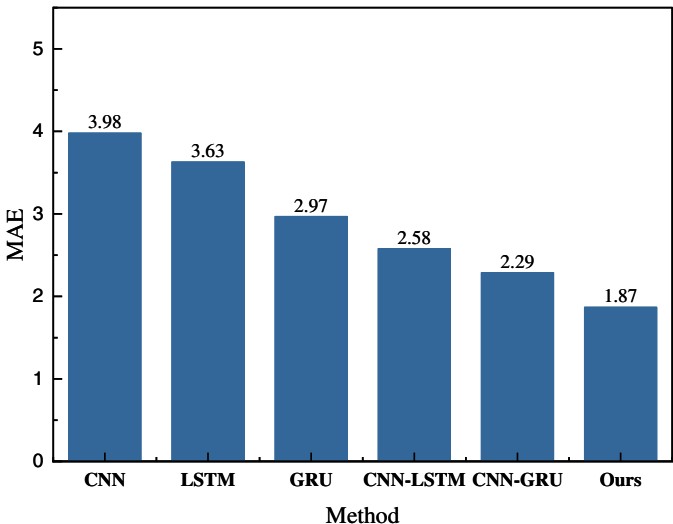
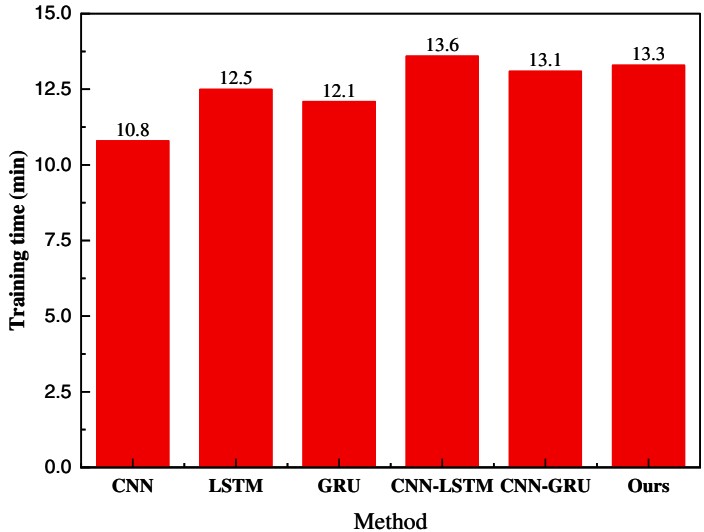

**Figure 5 MAE and training time on the public dataset.**

**Table 2 Training time.**

| Method | Training time (min) |
| --- | --- |
| CNN | 10.8 |
| LSTM | 12.5 |
| GRU | 12.1 |
| CNN-LSTM | 13.6 |
| CNN-GRU | 13.1 |
| Ours | 13.3 |

respectively, demonstrating an excellent fit for the smaller and less volatile price data. Building on this, we also predicted the production, with the results as follows:

Figure 8 shows that the prediction results for production and product price are similar, effectively fitting the data under fluctuating scenarios. The MSE and MAE for the prediction fitting are 19.01 and 15.33, respectively, which is favorable for production data with a mean value of around 1,000. To analyze the model's performance further under the cloud computing platform data stream, we also compared the MSE and MAE results of different methods, as shown in Fig. 9.

In Fig. 9, the proposed method outperforms other price and yield prediction approaches. This demonstrates that, in the context of cloud computing, our model effectively predicts the production and prices of agricultural products using multidimensional data. The performance is significantly better than that of single traditional methods, indicating a potential contribution to the advancement of the rural economy.

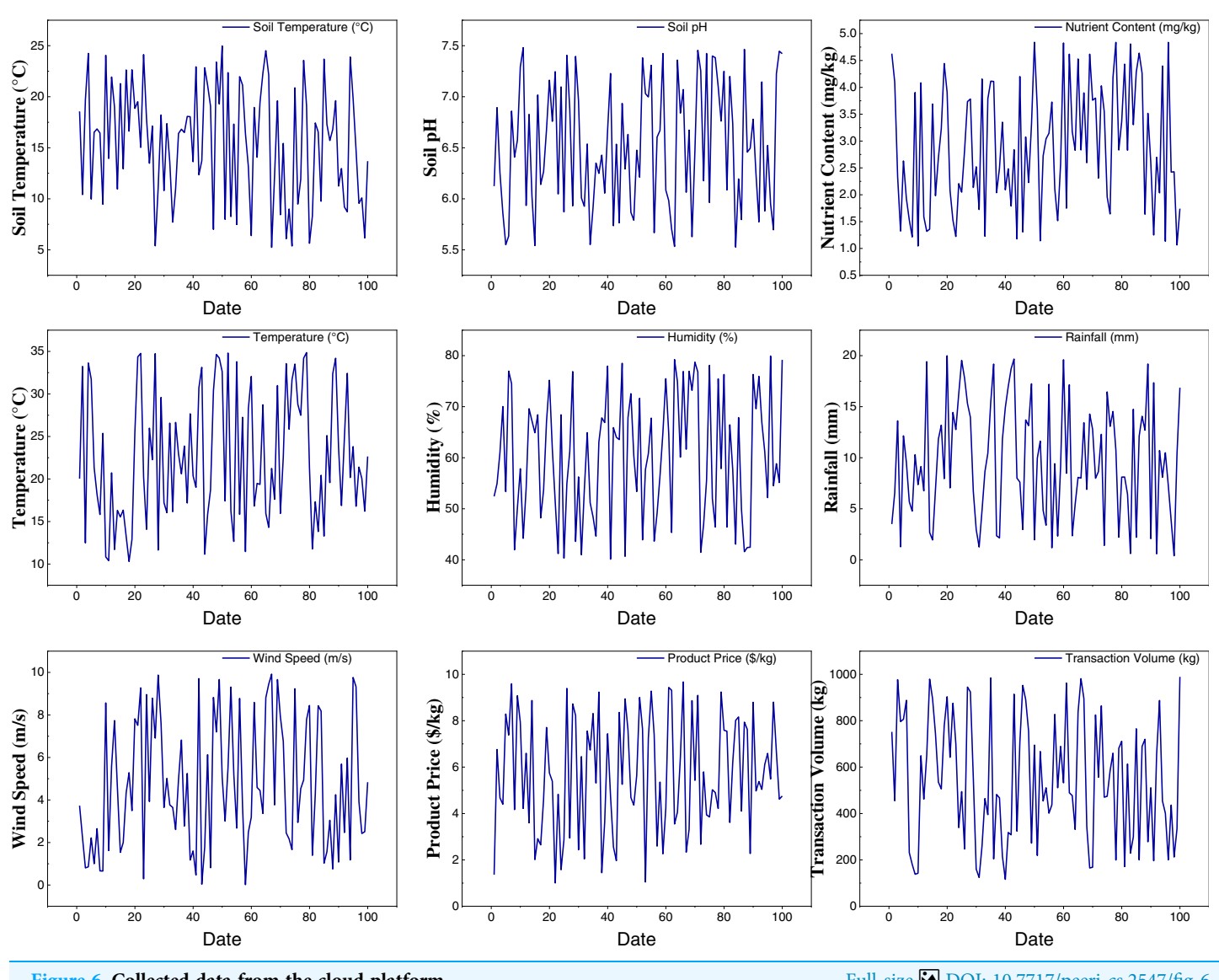

**Figure 6 Collected data from the cloud platform.**

## DISCUSSION

With the development of rural informatization and the application of big data technology, agricultural product price and yield prediction have become crucial means to enhance agricultural production efficiency and farmers' profitability. The RL-CNN-GRU framework proposed in this study demonstrates significant advantages in agricultural product price and yield prediction. Compared to traditional methods such as CNN, LSTM, GRU, and their combinations, the RL-CNN-GRU framework substantially improves the model's prediction accuracy by integrating multi-level feature extraction and reinforcement learning strategies. In this framework, the 1D convolutional neural network (1DCNN) serves as the first feature extraction layer, efficiently capturing patterns from historical price data and discrete factors through its convolutional and pooling layers. This

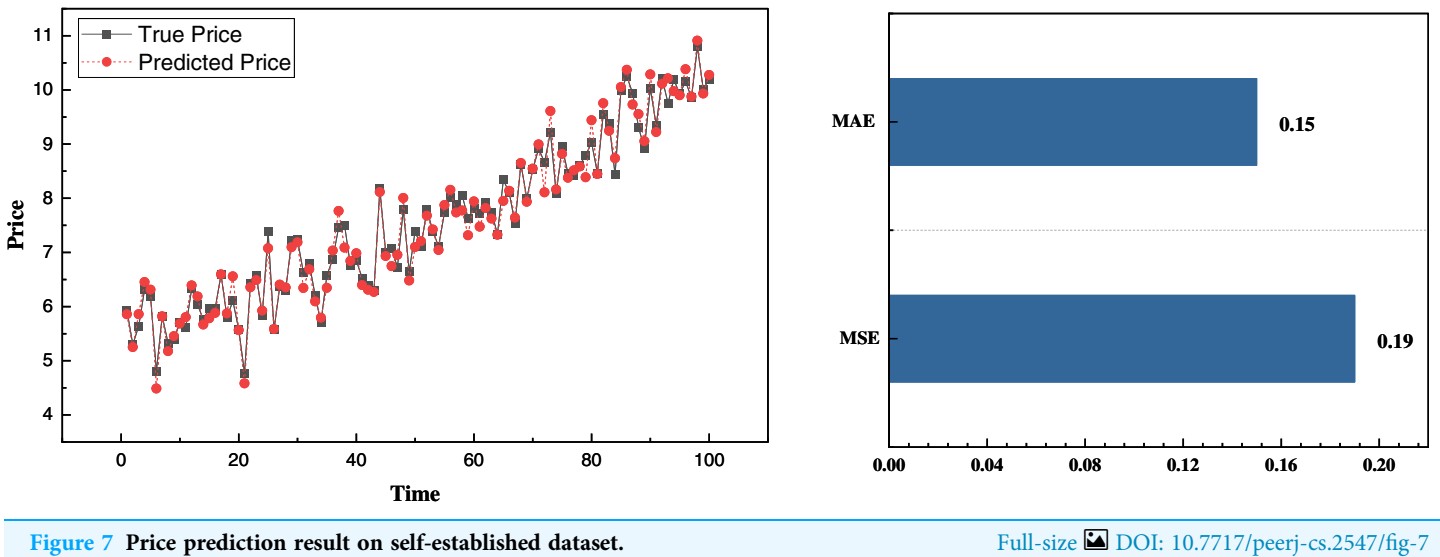

**Figure 7** Price prediction result on self-established dataset.

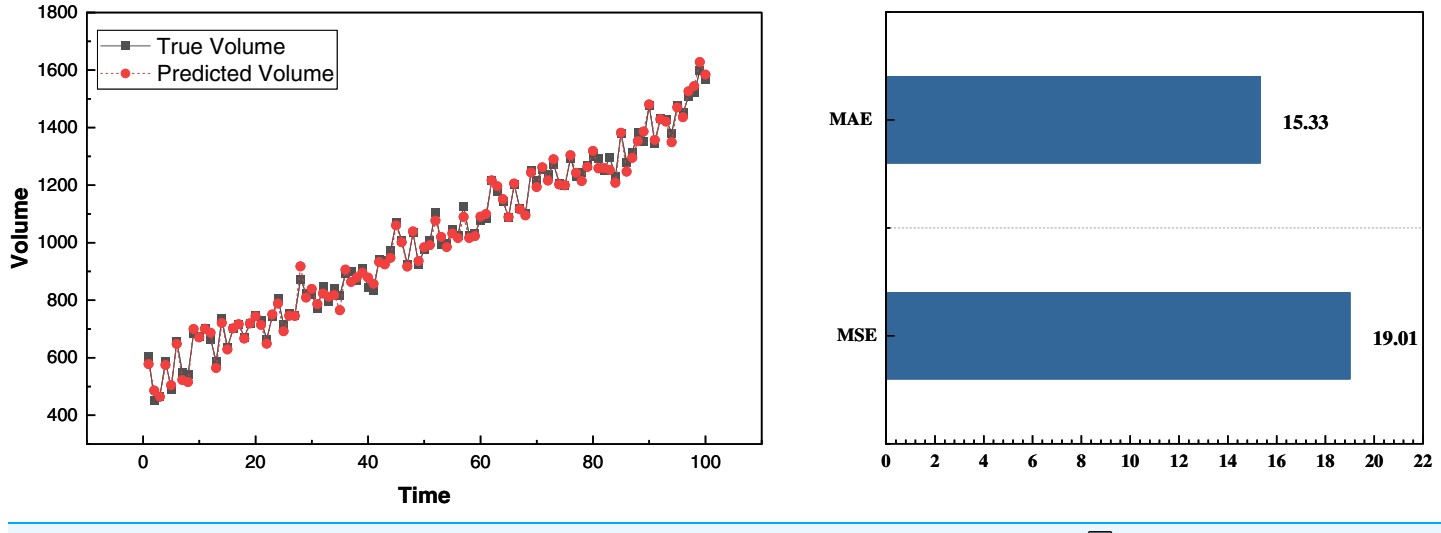

**Figure 8** Volume prediction result on self-established dataset.

architecture enhances the model's capacity to detect local dependencies and short-term trends in the time series data. Building on this, the GRU network is employed to further process the extracted features, improving the model's ability to capture both short- and long-term dependencies, which is essential for understanding the temporal dynamics of agricultural prices. Adding a reinforcement learning strategy significantly optimizes the model by dynamically adjusting to changing data patterns, improving the prediction accuracy and robustness of the price forecasting model. This combination of CNN, GRU, and reinforcement learning results in a robust, multi-layered framework capable of capturing complex nonlinear relationships and time dependencies more effectively than

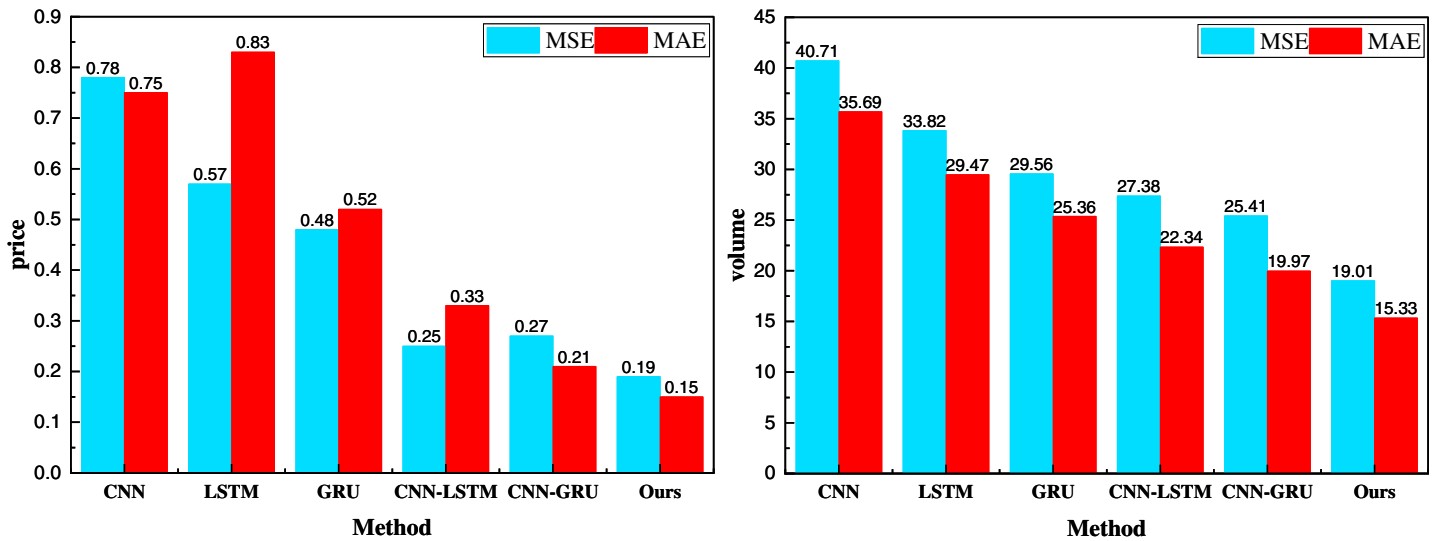

**Figure 9 MSE and MAE results for the method comparison.**               

standalone methods. Compared to using CNN or GRU alone, this hybrid model takes advantage of CNN's strength in local feature extraction and GRU's capability of handling temporal data, providing better performance in tasks with complex time-series data. Furthermore, while CNN-LSTM and CNN-GRU models can capture sequential dependencies, the integration of reinforcement learning adds an adaptive layer of optimization, allowing for continuous learning and improvement as new data becomes available. This ensures that the RL-CNN-GRU framework consistently outperforms traditional models like CNN, GRU, CNN-LSTM, and CNN-GRU, particularly in terms of reducing error metrics like MSE and MAE, making it more suitable for real-world agricultural forecasting where accuracy and adaptability are key.

With the continuous development of rural informatization and the maturity of cloud computing technology, this framework holds excellent significance for rural economic development. Firstly, accurate price and yield forecasts of agricultural products can assist farmers in devising more scientific planting and marketing strategies, thereby reducing market risks and enhancing economic returns. Secondly, the cloud computing platform plays a pivotal role in this study, enabling efficient collection and analysis of substantial agricultural data through cloud data processing and storage, thus providing a robust data foundation for model training and optimization. The flexibility and scalability of the cloud platform allow rural areas to quickly adapt to technological advancements, promoting the development of rural informatization. In future applications, it is crucial to focus on the data's quality and integrity to ensure the input data's accuracy and timeliness. The completeness of the data collected *via* cloud computing must be closely monitored. Additionally, model parameters and training strategies should be adjusted flexibly according to the specific conditions of agricultural production to leverage the model's predictive capabilities fully. In the future, with the incorporation of more real-time data

and sensor data, the RL-CNN-GRU framework is expected to improve prediction accuracy and application scope further. This will provide more substantial technical support for rural informatization, aiding the rural economy's comprehensive revitalization and sustainable development. Cloud technology enhances data analysis efficiency and scalability by delivering on-demand computing power and storage. For real-time data processing, cloud platforms offer distributed computing frameworks like Apache Spark, which can manage large-scale data streams efficiently. This capability ensures that models can ingest and analyze data continuously, enabling timely updates and actionable insights. Moreover, cloud services like AWS, Azure, and Google Cloud provide specialized tools such as auto-scaling and load balancing, which dynamically allocate resources to maintain performance even as data volumes fluctuate. However, potential issues include latency, especially when handling real-time updates across geographically dispersed cloud servers. Additionally, ensuring data security and maintaining compliance with privacy regulations (*e.g.*, GDPR) can be challenging, as data transmission over networks may expose sensitive information. Moreover, high computational demands and data storage costs can escalate, necessitating efficient resource management strategies to optimize cloud expenses. Cloud technology can only be better applied to efficient agricultural data analysis by solving the above problems.

## CONCLUSION

The RL-CNN-GRU framework based on reinforcement learning, CNN, and GRU proposed in this study provides an effective solution for agricultural price prediction in smart agriculture. By integrating CNN and GRU techniques with reinforcement learning strategies, we have constructed a prediction system capable of processing and analyzing multidimensional agricultural data. This approach successfully achieves deep modeling of multiple influencing factors, such as historical prices, meteorological conditions, soil conditions, and agricultural management, thereby significantly enhancing the model's prediction accuracy and reliability. The experimental results confirm the superior performance of the framework in practical applications, outperforming traditional methods like CNN, LSTM, and GRU and combined methods such as CNN-LSTM and CNN-GRU in terms of MSE and MAE. In the analysis based on the self-constructed dataset collected from the cloud computing platform, the RL-CNN-GRU framework also excels in practical application indices such as prediction accuracy and robustness. Its MSE indices for price and yield prediction are 0.19 and 19.01, respectively, nearly 10% better than the traditional optimal methods, demonstrating its potential in rural economic development. This research advances economic intelligence analysis and provides new technical means and strategic support for revitalizing the rural economy.

In future research, we plan to expand the framework's application by integrating additional meteorological, soil, and crop growth data to enhance its value in smart agriculture further. Specifically, we will utilize diverse data sources such as real-time weather stations, sensor networks, and satellite-based technologies like hyperspectral imaging for accurate and comprehensive data collection. Additionally, we aim to leverage advanced AI techniques, including meta-learning and adaptive algorithms, to improve the

framework's adaptability and generalization capabilities across different agricultural environments. These enhancements will strengthen the model's performance and offer more reliable technical support for agricultural production management and decision-making, driving sustainable development in smart agriculture.

## ACKNOWLEDGEMENTS

We thank the anonymous reviewers whose comments and suggestions helped to improve the manuscript.

### Funding

The authors received no funding for this work.

### Competing Interests

The authors declare that they have no competing interests.

### Author Contributions

- Haohao Chen conceived and designed the experiments, performed the experiments, analyzed the data, performed the computation work, prepared figures and/or tables, authored or reviewed drafts of the article, and approved the final draft.
- Fadi Al-Turjman conceived and designed the experiments, performed the experiments, analyzed the data, performed the computation work, prepared figures and/or tables, authored or reviewed drafts of the article, and approved the final draft.

### Data Availability

The code are available in the Supplemental File.

The drought dataset is available at Zenodo: Chuphal, D. S., Kushwaha, A. P., Aadhar, S., & Mishra, V. (2023). Drought Atlas of India, 1901–2020 [Data set]. Zenodo. https://doi.org/10.5281/zenodo.8280551.

### Supplemental Information

Supplemental information for this article can be found online at http://dx.doi.org/10.7717/peerj-cs.2547#supplemental-information.

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
