# Peer review of "Cloud-based configurable data stream processing architecture in rural economic development"

_PeerJ Computer Science, doi:10.7717/peerj-cs.2547_

## Round 0.1 · original submission · Major Revisions

Please see both reviewer's comments. Reviewers highlighted several areas for improvement, including a more detailed explanation of cloud platforms' performance in large-scale data processing, a clearer explanation of GRU mechanisms, and a comparison of the DDPG algorithm with other reinforcement learning strategies. Additionally, further details on hyperparameters, the model’s generalization ability across diverse data conditions, and a deeper analysis of experimental results are needed. Expanding on the methodology, discussing training time and scalability, and addressing performance under extreme conditions will enhance the paper’s impact.

·

Basic reporting

Introduction (Section 1, Paragraph 1): The background on "rural economic informatization" is too broad and does not sufficiently highlight the specific technological advancements or limitations in the field. It is recommended to expand the review of relevant literature on cutting-edge technologies in rural economic informatization, such as IoT, edge computing, and deep learning applications in agriculture, to provide a more comprehensive context.

Literature Review (Section 2, Paragraph 2): The discussion of cloud computing platforms' role in agricultural informatization lacks detailed analysis of real-world applications, particularly the practical integration of platforms like Amazon Web Services in agriculture. It is recommended to include detailed case studies that demonstrate the benefits of cloud platforms in data processing for smart agriculture, to strengthen the argument for their importance.

Experimental design

Methodology (Section 3, Equation 1): The mathematical representation of the convolution operation is unclear, especially the role of the convolutional kernel. This makes it difficult for readers to understand its application to time-series data processing. It is recommended to add an explanation after Equation (1) detailing the function of the convolutional kernel and time step, along with a diagram to help readers visualize the process.
Methodology (Section 3, Paragraph 2, CNN and GRU combination): The rationale for combining CNN and GRU is insufficiently explained. More detailed justification for choosing 1D-CNN over other types of convolutional layers is needed, especially when compared to other models for time-series data such as Transformer or attention-based mechanisms. Experiments comparing their performance could provide a stronger theoretical foundation.

Validity of the findings

Results (Section 4, Figures 4 and 5): While Figures 4 and 5 show the loss function and MAE comparisons, there is no discussion about the impact of training time. Given the importance of time complexity and computational resources in cloud environments, it is recommended to add a discussion on training time and complexity, and compare the time cost between different models through charts.

Results (Section 4, Table 1): Table 1 does not provide detailed information about the dataset size and training time. Considering the large-scale data processing typical in cloud computing environments, it is recommended to include more details on training time and resource consumption at different dataset scales to clarify the model's scalability.

Discussion (Section 5, Paragraph 1): The current analysis of the model's suitability for cloud environments is not deep enough. The paper does not address how the model handles real-time data updates in the cloud. It is recommended to expand the discussion on the model's performance with real-time, large-scale data and analyze the potential bottlenecks in cloud data processing.

Discussion (Section 5, Figures 7 and 8): Figures 7 and 8 show the prediction results for price and production but fail to address the model's performance under extreme conditions, such as during periods of significant data volatility or outliers. It is recommended to include experimental results under such scenarios to assess the model’s robustness in complex environments and its reliability in real-world applications.

Reviewer 2 ·

Basic reporting

.

Experimental design

.

Validity of the findings

.

Additional comments

This paper introduces an innovative framework combining reinforcement learning and deep learning in agricultural product price forecasting, and performs data processing through cloud computing platform, which has high application potential. However, there is still much room for improvement in the technical implementation and experimental design, especially in the details of the method and the analysis of the results.

 The literature review mentions cloud platforms’ support for agricultural informatization but does not explain their performance in large-scale data processing or forecasting. It is recommended to add more detail about these platforms’ capabilities and limitations when handling multi-source, large-scale agricultural data, particularly in real-world applications, to enhance the depth of the review.

 The explanation of the GRU mathematical formulas is insufficient, particularly concerning the gate mechanism. It is recommended to add a detailed explanation of the update and reset gates in GRU, specifically how they function in long-sequence data processing, and provide a flowchart to better illustrate the GRU’s process.

 The paper uses the DDPG algorithm but does not compare it with other reinforcement learning strategies such as PPO or SAC, making it hard to justify its superiority. It is recommended to include experiments or theoretical comparisons to demonstrate why DDPG was chosen and how it performs compared to alternative strategies in agricultural data prediction.

 The experimental results do not provide enough detail about the hyperparameters used in different models, particularly the choice of convolution kernel size, learning rate, and other key parameters. It is recommended to include more details about these hyperparameters and their influence on model performance to make the optimization process more transparent.

 The discussion focuses on the model’s advantage in price prediction but does not elaborate on its generalization ability across different agricultural data conditions. It is recommended to include experiments with data from varying climates or crop types to demonstrate the model’s robustness and wide applicability.

 Table 3 shows MSE and MAE values across different models, but there is little in-depth analysis of these results. It is recommended to provide a comparison of how each model handles large-scale or outlier data, as well as an analysis of each method’s strengths and weaknesses, to better explain the advantages of the proposed method.

 The conclusion mentions that more meteorological and soil data will be incorporated in future work but does not specify the technical approach or data sources. It is recommended to detail the types of data that will be used, how they will be collected, and whether new technologies such as sensor networks or hyperspectral imaging will be employed to provide a clearer future direction for the research.

---

## Round 0.2 · accepted · Accept

Both reviewers have confirmed their comments are now addressed.

·

Basic reporting

The writer accommodated fully all the recommendation

Experimental design

it is well satisfied according to the research article

Validity of the findings

all attributes are valid

Reviewer 2 ·

Basic reporting

This is revision. All my concerns have been reflect.

Experimental design

.

Validity of the findings

.

Additional comments

.